# Self-supervised Learning to Discover Physical Objects and Predict Their Interactions from Raw Videos

## Abstract

The ability to discover objects from raw videos and to predict their future dynamics is crucial for achieving general intelligence. While existing methods accomplish these two tasks separately, i.e., learning object segmentation with fixed dynamics or learning dynamics with known system states, we explore the feasibility of jointly accomplishing the two together in a self-supervised setting for physical environments. Critically, we show on real video datasets that learning object dynamics improves the accuracy of discovering dynamical objects.

## 1 Introduction

Cognitive science researchers have studied how humans understand both scenes and events since the 1970s [3, 8, 51, 45]. Inspired by these studies, AI researchers have been striving to build intelligence systems with similar abilities [59, 5, 60, 12]. Most recent work pursues these two objectives *separately*, e.g., supervised and unsupervised object discovery [28, 32, 55, 39, 26, 25, 7, 41, 20] and learning physics and dynamics from data [9, 54, 55].

Meanwhile, inspired by cognitive science research about how infants can develop their perceptual system and learn the physical world simultaneously in a self-supervised fashion by observing and interacting with moving objects [33, 2], recent studies hypothesize that such joint learning of object discovery and dynamics should also be feasible for machines. In particular, recent progress on object discovery from motion, e.g., [52, 53, 18, 56, 50, 36, 19], shows that the *existence* of dynamics prediction, even when the dynamical models are primitive, improves the accuracy of object discovery.

In parallel, machine learning for physics has achieved significant progress in recent years, with applications to physical property prediction [24], protein or material generation [34, 14, 47], particle-based simulation [46, 11], among many others. Notably, neural ODE [11] and its successor [27, 15, 43] have demonstrated strong capabilities of neural networks in approximating dynamical systems. In most settings, however, the states of physical objects are assumed to be given, with only a few studies, e.g., [9, 18], attempted to learn state of objects from video.

In this work, we show that the accuracy of object discovery in physical environments can be further improved when the dynamical model is trainable and represents a hypothesis space that covers the ground truth dynamics, and on the other hand, an incorrect assumption about dynamics may result in faulty segmentation of objects. As shown in Figure 1, our model is based on a factorized generative model for object discovery and a trainable neural ODE for dynamics prediction [11]. The component linking the object discovery and the dynamical model is a state encoder, which maps a time sequence of object masks to object states such as position, orientation, and their time derivatives. Unlike

Submitted to NeurIPS 2021 AI for Science Workshop.

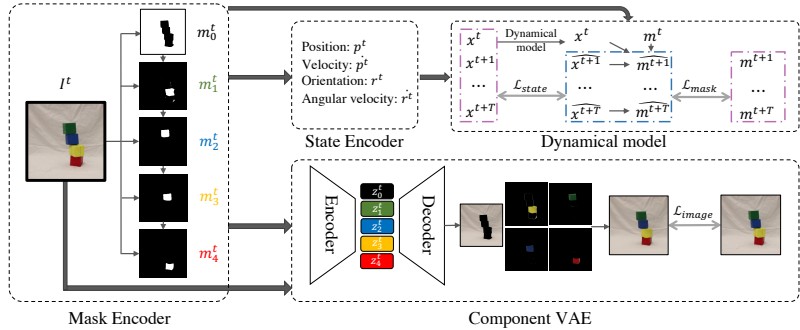

Figure 1: Our framework consists of four components: mask encoder, component VAE, state encoder, and dynamical model.

previous studies where the dynamics is fixed, our model introduces the challenge of jointly learning for object discovery and for dynamics prediction.

The key contributions of the paper are as follows:

- We present effective learning architecture, loss, and algorithm for solving the challenge posed by the joint learning task.
- We empirically test our model on two video datasets: real-world double pendulum, and real-world 3D block tower falling. We show that through joint learning of object discovery and dynamical model, our method outperforms recent object segmentation methods that use factorized generative model [7, 41] or primitive dynamics [18, 53]. The learned dynamical model can also predict the movement of objects in long-term.

## 2   Method

By integrating a trainable dynamical model into an object discovery framework, our model jointly learns object masks and predicts object interactions. As shown in Figure 1, The learning framework is composed of: (1) a mask encoder that encodes an input video frame into object masks using attention modules, (2) a component variational autoencoder (VAE) that encodes the concatenated image and object masks into object-wise latent representations, which can be decoded back into an image and reconstructed masks, (3) a prefixed state encoder that computes the center of mass, orientations, and their time derivatives for each masked object, (4) a dynamical model that evolves states along time.

**The mask encoder.** Let a video with $T$ time frames be $\mathcal{I} = \{I^0, ..., I^T\}$, where $I^t \in \mathbb{R}^{H \times W \times 3}$ is an RGB image with height $H$ and width $W$. A mask encoder, denoted by $f_\psi(\cdot)$ with trainable parameters $\psi$, encodes an image into one background mask and $C$ object masks: $f_\psi(I^t) = \mathcal{M}^t \triangleq \{m_0^t, m_1^t, ..., m_C^t\}$, where $m_c^t \in [0,1]^{H \times W}$ and $m_0^t$ represents the background mask. Since masks should cover all pixels in the scene, the sum of all masks is 1: $\sum_{c=0}^{C} m_c^t = J_{H,W}$. Let $q_c$ represent the area unexplored until iteration $c$. To discover objects in the scene, we adopt the method in [7]. The attention module $\text{Attention}_\psi$ recurrently discovers objects through

$$m_c = q_{c-1}(\text{Attention}_\psi(I, q_{c-1})), q_c = q_{c-1}(1 - \text{Attention}_\psi(I, q_{c-1})), \ \forall c = 1, ..., C, q_0 = \mathbf{1}. \ (1)$$

**The component VAE.** For the $c^{th}$ mask, the encoder encodes the image $I$ to a latent posterior distribution, denoted as $p_\phi(z_c|I, m_c)$. The latent vector $z_c$ for each mask $m_c$ is decoded back to both the image likelihood $p_\theta(I_c|z_c)$ and the mask prediction likelihood $p_\theta(d_c|z_c)$. The reconstructed image is a summation over all channels $I = \sum_{c=0}^{C} m_c I_c$. $\phi$ and $\theta$ are trainable component VAE encoder and decoder parameters.

**The state encoder.** computes the state $(x_c^t)$ based on each object mask $(m_c^t)$: $f_s(m_c^t) = x_c^t$. The state is composed of the center of mass $p_c^t \in \mathbb{R}^2$, velocity $\dot{p}_c^t \in \mathbb{R}^2$, orientation $r_c^t \in \mathbb{R}$, and angular velocity $\dot{r}_c^t \in \mathbb{R}$ of each object. Therefore $x_c^t \in \mathbb{R}^6$. In our implementation, the state encoder first extracts pixel coordinates of an object based on its mask and then computes the state from these

coordinates. The collection of coordinates $l_c^t$ is computed by an element-wise multiplication of mask $m_c^t$ with a 2D coordinate grid $g \in [-1,1]^{H \times W \times 2}$. The center of mass $p_c^t$ is retrieved as the mean of $l_c^t$, and the orientation $r_c^t$ as the direction of the principle axis of $l_c^t$ through differentiable singular value decomposition. The time derivatives $\dot{p}_c^t$ and $\dot{r}_c^t$ are computed by a finite difference using the position and orientation of the current and the previous time steps: $\dot{p}_c^t = p_c^t - p_c^{t-1}, \dot{r}_c^t = r_c^t - r_c^{t-1}$. Note that the state encoder is a non-trainable differentiable program, which is able to backpropagate gradients from the dynamical model back to the mask encoder.

**The dynamical model.** The dynamical model $f_\xi$ predicts future states given the current state: $x^{t+\Delta t} = f_\xi(x^t, \Delta t)$, where $\xi$ are trainable model parameters and $\Delta t$ is a time span. The state $x^t$ is concatenated by states of each mask, denoted as $x^t = [x_1^t, ..., x_C^t] \in \mathbb{R}^{C \times 6}$. $f_\xi$ is composed of a neural ODE: $\dot{x}^t = f_{ode}(x^t, \Delta t)$, and a differentiable ODE solver (e.g., Euler or Runge-Kutta):

$$\widehat{x^{t+\Delta t}} = f_\xi(x^t, \Delta t) = \text{ODESolver}(f_{ode}, x^t, t, t + \Delta t). \tag{2}$$

Future object masks can be predicted by applying affine transformations using the predicted states:

$$\widehat{m_c^{t+1}} = \tau(\widehat{m_c^t}, \widehat{x_c^{t+1}}, \widehat{x_c^t}), \widehat{m_c^0} = m_c^0, \widehat{x_c^0} = x_c^0, \forall c \geq 1, \tag{3}$$

where $\tau$ is a differentiable affine transformation given rotation and translation [31].

**Training losses.** In a nutshell, the training loss consists of (1) a time-independent reconstruction loss regarding the mask encoder and the VAE, and (2) a time-dependent dynamics loss that is dependent on both the mask encoder and the dynamical model. The reconstruction loss includes a standard VAE loss and a KL regularization. The VAE loss has two terms: the first term is the negative log-likelihood of the generated image distribution, denoted as $\mathcal{L}_\theta = -\log \sum_{c=0}^{C} m_c p_\theta(I|z_c)$; the second term is the KL divergence of the learned latent distribution from the prior, denoted as $\mathcal{L}_\phi = KL(p_\phi(z_c|I, m_c)||p(z))$, where the prior follows a standard normal distribution: $p(z) = \mathcal{N}(0, 1)$. The KL regularization loss is the KL divergence of the encoded mask distribution from the decoded mask prediction distribution, denoted as $\mathcal{L}_{\psi,\theta} = KL(p_\psi(d_c|I)||p_\theta(m_c|z_c))$. Together, the reconstruction loss is:

$$\mathcal{L}_{recon} = \min_{\psi,\phi,\theta} \mathcal{L}_\theta + \alpha \mathcal{L}_{\psi,\theta} + \beta \mathcal{L}_{\psi,\theta} \tag{4}$$

The dynamics loss is composed of a state loss and a mask loss. The state loss measures the difference between the state encoded from an image and the state predicted by the dynamical model using past encoded states. Through preliminary experiments, we notice that the state loss alone may lead to a trivial solution during training convergence, where states are both encoded and predicted as being constant, therefore minimizing the state loss without learning the actual dynamics. To avoid this, we introduce an additional mask loss that measures the difference between the masks encoded from the image and those evolved by the dynamical model. Thus, the performance of dynamics prediction is measured in both the state and the mask spaces. Together, the dynamics loss is:

$$\mathcal{L}_{dynamics} = \min_{\psi,\xi} \sum_{t=1}^{T} \left( \left\| \widehat{x^t} - x^t \right\|_2 + \gamma \sum_{c=1}^{C} \left\| \widehat{m_c^t} - m_c^t \right\|_2 \right). \tag{5}$$

The overall training loss is a weighted sum of the reconstruction, dynamics, and regularization loss:

$$\mathcal{L}_{total} = \mathcal{L}_{recon} + \eta \mathcal{L}_{dynamics}. \tag{6}$$

# 3 Experiments

**Experiment settings.** We conduct experiments on video datasets of two physical environments: a video-recorded double pendulum dataset, and a video-recorded 3D block tower dataset. The double-pendulum dataset is video recorded from actual experiments and shared by [9]. The 3D block tower dataset, introduced in [38], provides a collection of videos showcasing block stacks that may or may not fall. The dataset comprises 516 videos, each featuring 2 to 4 blocks of various colors. To quantify the object discovery performance, we employ the intersection over union (IoU) metric, which compares the encoded masks and ground truth segmentation.

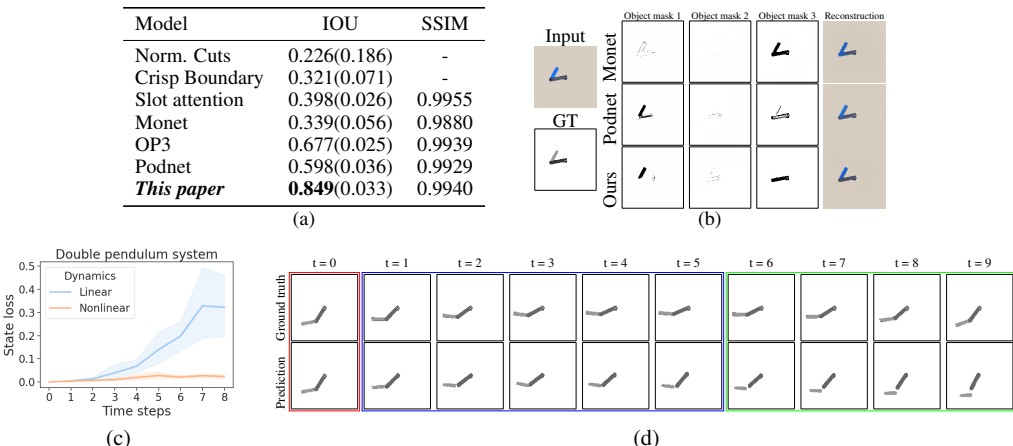

| Model | IOU | SSIM |
|---|---|---|
| Norm. Cuts | 0.226(0.186) | - |
| Crisp Boundary | 0.321(0.071) | - |
| Slot attention | 0.398(0.026) | 0.9955 |
| Monet | 0.339(0.056) | 0.9880 |
| OP3 | 0.677(0.025) | 0.9939 |
| Podnet | 0.598(0.036) | 0.9929 |
| *This paper* | **0.849**(0.033) | 0.9940 |

(a)

(b)

(c)

(d)

Figure 2: The quantitative and qualitative object discovery and dynamical prediction result on the double pendulum dataset. (a) The quantitative object discovery result; (b) The qualitative object discovery result; (c) The state loss over time; (d) The qualitative dynamical prediction results.

**Real-world video recording of double pendulum.** We compare our method against baselines in Figure 2(a). Our method performs the best, with only a few pixels on the edge of the blue pendulum being mistakenly grouped with the gray pendulum, as shown in Figure 2(b). We note that while our model achieves low dynamics prediction error in the state space (Figure 2(c)), it has limited understanding of geometric relations of objects (Figure 2(d)), leaving room for improvement.

**3D Real-world Block Tower.** To compute 3D states from 2D masks, we first extract 2D states from our state encoder and then project them to 3D using the back-projection model pretrained by [18]. Since the number of objects in the scene can vary, we choose to measure the detection performance of models as well as the object segmentation IoU for evaluation.

We compare our method with baselines in Figure 3. We observe that Monet tends to group objects with similar colors together, such as the blue and green blocks, and occasionally misclassifies light or dark regions as part of the background. Podnet exhibits good object detection performance but encounters challenges in object discovery, as shown in Figure 3. Podnet struggles with accurately delineating the boundary between the green and blue blocks. In the second row, it misidentifies a shadow as an object rather than perceiving it as part of the background. Additionally, in the third row, it fails to detect a portion of the yellow block. In comparison, our model achieves more

| Model | IoU | Detection |
|---|---|---|
| Norm. Cuts | 0.652 (0.006) | 0.849 (0.018) |
| UVOD | 0.029 (0.001) | 0.0 (0.0) |
| Monet | 0.521 (0.005) | 0.537 (0.003) |
| OP3 | 0.311 (0.004) | 0.250 (0.007) |
| Podnet | 0.837 (0.004) | 0.908 (0.008) |
| *This paper* | **0.898** (0.016) | **1.0**(0.0) |

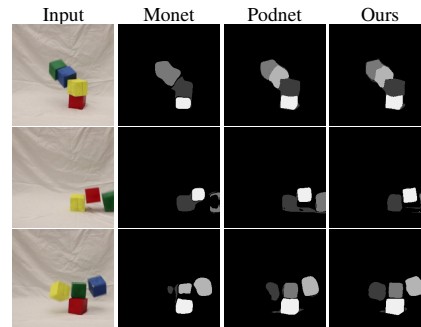

Figure 3: The quantitative and qualitative object discovery on the block tower dataset.

accurate object detection and object discovery. This improvement highlights the effectiveness of incorporating a trainable nonlinear dynamical model into the segmentation framework.

## 4 Conclusion

In this work, we present a model that decomposes images into multiple objects and predicts the dynamics of these objects. We show that ill-posed assumptions of dynamics may result in false object discovery. Our model with trainable nonlinear dynamics is capable of accurately discovering objects while predicting their future movements. For future work, we envision an extension to interactions among non-rigid objects that require both explicit and implicit state encoding for time-variant shape and color changes (e.g., cell migration).

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

## A  Related Work

**Object discovery from static images.** Our method is related to object discovery, which aims to decompose a scene into compositional objects by segmentation. Motivated by Gestalt psychology [37], object discovery enables scene understanding [22, 59, 60, 40], vision reasoning [17, 59], and physical reasoning [13, 55, 12, 54, 5, 29]. The conventional approach to object discovery is to cluster the image pixels based on low-level vision information such as texture and color using graph-based inference [44] or normalized cuts algorithms [21]. Learning-based methods often require supervisory information such as segmentation masks [28, 32], physical simulators [55, 39], or depth maps [19]. However, such supervisory data can be expensive or sometimes infeasible. Recent self-supervised methods learn to discover objects by minimizing a reconstruction loss, i.e., they encode scenes into

masks, which are then decoded back to scenes. [26, 25, 7, 41, 20]. Among these, [7, 20] discover objects one-by-one during the encoding, and [41, 26] discover all objects together but iteratively refine the discovery.

**Motion segmentation.** Motion segmentation extends object discovery from static images to videos; yet, it is conventionally less concerned about predicting future movements of the discovered objects. Optical flow is a conventional method to segment all moving objects as foreground of videos [6, 35]. Recent learning based approaches rely on salient motions to segment common objects with similar appearance from video [56, 16, 23, 4, 42, 57, 10, 49, 58, 1].

**Learning dynamics for physical environment.** In parallel to motion segmentation are studies on learning dynamics, which focus on training dynamical models to predict system states, while assuming that the definition of system states are known. Recent work attempts to directly learn state representations and dynamics through images. Among these, [9] estimates the dimension of the latent state space via intrinsic dimension estimation. Similar to these efforts, our method jointly learns state representation and dynamics, but instead of learning a latent representation which has unknown physical meaning, we explicitly encode states as object center of mass and orientation, which are *interpretable* and suffice for rigid objects. Extension to soft bodies is possible, but will be left for future work.

**Object discovery using dynamics.** Our study is most relevant to object discovery using dynamics, where objects and their dynamics are jointly learned from raw videos [52, 53, 18, 56, 50, 36, 19]. The key idea is that both the dynamics that govern the interaction of objects and some object properties, e.g., geometries of rigid bodies, are time-invariant and can be used as an inductive bias to improve the learning of object discovery. Among these studies, [18] learns object states and predicts their future states using linear extrapolation. [53] discovers entity variables by a model-base reinforcement learner. [52] segments the objects by modeling the relations and interaction of objects using a recurrent neural network. These existing studies use simple and fixed dynamical models to support object discovery. We show in this paper that the accuracy of object discovery can be further improved by jointly learning a dynamical model from a hypothesis space that covers the true dynamics.

# B    Detailed Experiment Setting

**Experiment setup.**    The architecture of the mask encoder and the VAE follows that of Monet [7]. In experiments, we set the number of object masks to $C = 3$. Before we compute object states from the masks, we filter out masks with less than 5 activated pixels with the assumption that small objects do not exist (or should not affect the dynamics). This treatment helps the convergence. Also note that the computation of the principal axis is direction agnostic because both $v$ and $-v$ are eigenvectors of a data matrix. Therefore instead of computing the angle and angular velocity $(r, \frac{dr}{dt})$, we compute $(\cos^2 r, \frac{d\cos^2 r}{dt})$ which have a period of $\pi$, and use these in the computation of state losses. The $(r, dr)$ are still used in the affine transformation function to compute the mask losses.

The trainable dynamical model for the two-body system is a four-layer fully-connected feedforward network with 20 neurons for each hidden layer. For the double pendulum case, we expand the network to 5 layers, with $[20, 40, 40, 20]$ neurons for the respective hidden layers. For the block tower dataset, we use the same dynamical model as double pendulum. We use `tanh` as the activation for all networks. In our experiment, the length of dynamics for training $T$ is 5.

The training process consists of two steps. Following Podnet, we first pre-train the mask encoder and the VAE to minimize the reconstruction loss until convergence. This is because adding dynamics loss at an early stage when no objects are discovered and states are physically meaningless will destabilize the training process. After the pretrain stage, the mask encoder can successfully separate objects out from the background, although multiple objects can still be mistakenly grouped as one. Next, we train the whole model including dynamical model to jointly minimize the reconstruction and

the dynamics losses. The hyperparameters are set to $\alpha = 0.5$, $\beta = 0.25$, $\gamma = 1$ and $\eta = 1e^4$. The optimizer is RMSprop and the learning rate is 1e-4. We use a NIVIDA-V100 for all training.

**Baseline.** Two conventional algorithms for unsupervised object segmentation: normalized cuts [48], and crispy boundary detection [30], as well as four learning-based unsupervised/self-supervised object discovery methods: Monet [7], Slot attention [41], Podnet [18], and OP3 [53], are used as baselines for comparison. Normalized cuts is a graph partition method treating pixels of an image as vertices of a graph, partitioning groups of vertices measured by normalized cut. Crisp boundary detection is a semantic edge detection method and can also be used for image segmentation by edges. Monet and Slot attention are unsupervised encoder and decoder architecture, but do not leverage dynamics. As an improvement from Monet, Podnet uses dynamics for object discovery, yet the non-trainable dynamics follows simple linear extrapolation: $x^t = f_\xi(x^{t-1}) = x^{t-1} + \frac{1}{t-1}\sum_{i=1}^{t-1}(x^i - x^{i-1})$. OP3 uses a probabilistic dynamical model on the object-centric latent variable to discover the objects. The details of baseline setup are in the appendix. Our method is different from Podnet in that we introduce a trainable dynamical model that is flexible enough to cover the ground truth dynamics instead of linear extrapolation. Compared with OP3, the state in our model has known physical meaning and the dynamical model is deterministic.

**Performance metrics.** To quantify the object discovery performance, we employ the intersection over union (IoU) metric, which compares the encoded masks and ground truth segmentation. Since the encoded masks (i.e., the three channels) can be ordered differently than the ground truth, we compute IoU by pairing a ground truth segmentation with each encoded mask and take the maximum IoU. We then take the average IoU across all test video frames. Additionally, for most learning-based methods, we incorporate the assessment of image reconstruction quality using the structural similarity index measure (SSIM), reflecting the convergence of the training algorithm. It is important to note that even if the learning achieves high image reconstruction quality, the learned model may still not be proficient at correctly identifying objects if the performance metric for object discovery is low.

To quantify the dynamics prediction performance, we report the error between states computed from the masks and those predicted by the dynamical model. In addition, we visualize the evolution of object masks by computing the masks from the mask encoder for the initial frame and applying affine transformations based on the predicted states for up to 9 time steps, as described in Equation 3. The time derivatives $\dot{p}_c^0$ and $\dot{r}_c^0$ are computed by the first two video frames along with the mask encoder.

