# OpenReview forum: "Self-supervised Learning to Discover Physical Objects and Predict Their Interactions from Raw Videos"
_NeurIPS.cc/2023/Workshop/AI4Science — NeurIPS2023-AI4Science Poster_

### Official Review · Reviewer_gJi5 · 2023-10-15

**Rating:** 7
**Confidence:** 3

**Review:**

**Summary & Strengths**

The paper introduces a novel approach that concurrently learns object dynamics and object discovery from input video pixels. The method proposes segmenting objects and learning the physical dynamics of each, utilizing a mask encoder to apprehend object-centric representations. Subsequently, it calculates the physical states of the objects, employing an ODE solver to predict the objects' states in the ensuing time step. The approach demonstrates commendable performance on two video datasets and presents a coherent, straightforward scheme with the potential to benefit various downstream fields such as AI for Science and MBRL. I recommend acceptance based on the potential and clarity of the method.

**Questions & Suggestions**

- *Use of Reconstruction Loss*: The paper opts for reconstruction loss in its methodology. Could the authors elaborate on the decision not to incorporate contrastive loss [1], given its potential for superior learning in temporal and multi-object discovery contexts?

[1] Kipf, Thomas, Elise Van der Pol, and Max Welling. "Contrastive learning of structured world models." ICLR 2020.

- *Applicability in Downstream Tasks:* Can the module provide benefits for downstream tasks, especially in MBRL, particularly when video is utilized as input? Is it plausible that learning a robust representation and dynamics model could enhance RL performance?

- *Multi-Step Prediction*: In scenarios that involve multi-step prediction, could the authors provide empirical results regarding performance versus the number of steps?

- *Real-World Scenario Applicability*: How well does the method adapt to more realistic scenarios, such as those involving interactions between multiple objects? Could integrating a prior or additional modules, such as Neural Relational Inference (NRI) [2], at the dynamics model stage be considered to facilitate this?

[2] Kipf, Thomas, et al. "Neural relational inference for interacting systems." International conference on machine learning. PMLR, 2018.

- *Complex, Real-World Dataset Consideration*: Is there scope for the method to be evaluated against more complex and realistic datasets, for instance, those pertaining to embodied AI (some multi-object dynamics datasets in link [3]) and multi-object interactive environments [4]?

[3] https://embodied-ai.org/

[4] Yang, Mengjiao, et al. "Learning Interactive Real-World Simulators." arXiv preprint arXiv:2310.06114 (2023).

---

### Meta-Review · Area_Chair_wjLU · 2023-10-27

**Recommendation:** Accept (Oral)
**Confidence:** 4

**Metareview:**

In this paper, authors claim to propose first work to jointly learn to discover physical objects and predict their dynamics in the videos, which helps to improve the results. Overall, the paper is clearly written and presents sufficient experiments to demonstrate the effetiveness of the idea. I agree with reviewer to recommend this work due to its novel approach.